# Roadmap for Stroke: Challenging the Role of the Neuronal Extracellular Matrix

**DOI:** 10.3390/ijms21207554

**Published:** 2020-10-13

**Authors:** Ciro De Luca, Assunta Virtuoso, Nicola Maggio, Sara Izzo, Michele Papa, Anna Maria Colangelo

**Affiliations:** 1Laboratory of Morphology of Neuronal Network, Department of Public Medicine, University of Campania “Luigi Vanvitelli”, 80138 Napoli, Italy; ciro.deluca@unicampania.it (C.D.L.); assunta.virtuoso@unicampania.it (A.V.); 2School of Medicine and Surgery, University of Milano-Bicocca, 20900 Monza, Italy; 3Department of Neurology and Neurosurgery, Sackler Faculty of Medicine, Sagol School of Neuroscience, Tel Aviv University, Tel Aviv 6997801, Israel; Nicola.maggio@sheba.health.gov.il; 4Department of Neurology and the J. Sagol Neuroscience Center, The Chaim Sheba Medical Center, Tel HaShomer, Ramat Gan 5211401, Israel; 5Multidisciplinary Department of Medical-Surgical and Dental Specialties, Plastic Surgery Unit, University of Campania “Luigi Vanvitelli”, 80138 Napoli, Italy; sa_izzo@hotmail.it; 6SYSBIO Centre of Systems Biology ISBE.ITALY, University of Milano-Bicocca, 20126 Milano, Italy; annamaria.colangelo@unimib.it; 7Laboratory of Neuroscience “R. Levi-Montalcini”, Department of Biotechnology and Biosciences, University of Milano-Bicocca, 20126 Milano, Italy

**Keywords:** stroke, neuronal extracellular matrix, neurovascular unit, matrix metalloproteinases, systems biology, maladaptive plasticity

## Abstract

Stroke is a major challenge in modern medicine and understanding the role of the neuronal extracellular matrix (NECM) in its pathophysiology is fundamental for promoting brain repair. Currently, stroke research is focused on the neurovascular unit (NVU). Impairment of the NVU leads to neuronal loss through post-ischemic and reperfusion injuries, as well as coagulatory and inflammatory processes. The ictal core is produced in a few minutes by the high metabolic demand of the central nervous system. Uncontrolled or prolonged inflammatory response is characterized by leukocyte infiltration of the injured site that is limited by astroglial reaction. The metabolic failure reshapes the NECM through matrix metalloproteinases (MMPs) and novel deposition of structural proteins continues within months of the acute event. These maladaptive reparative processes are responsible for the neurological clinical phenotype. In this review, we aim to provide a systems biology approach to stroke pathophysiology, relating the injury to the NVU with the pervasive metabolic failure, inflammatory response and modifications of the NECM. The available data will be used to build a protein–protein interaction (PPI) map starting with 38 proteins involved in stroke pathophysiology, taking into account the timeline of damage and the co-expression scores of their RNA patterns The application of the proposed network could lead to a more accurate design of translational experiments aiming at improving both the therapy and the rehabilitation processes.

## 1. Introduction

As life expectancy increases, vascular diseases are progressively rising as major causes of disability and death [1]. The central nervous system (CNS) is no exception and stroke represents one of the most common and challenging pathologies of recent decades [2]. Stroke, along with neurodegeneration, can have a great impact on personal health and social burden for elderly people. Indeed, the increasing prevalence of neurovascular disorders, alone or in combination with degenerative diseases, dramatically enhances so-called vascular cognitive impairment (VCI) [3].

Various definitions of stroke have been assessed, however, the American Heart Association/American Stroke Association (AHA/ASA) delineated stroke as a CNS infarction within tissues of the brain, spinal cord or retina, due to ischemia [4]. It should be confirmed through radiological, clinical and/or pathological evidence of permanent injury. Inside the AHA/ASA statement are ischemic or hemorrhagic stroke and subarachnoid hemorrhage. The central element to unravel the vascular disease physiopathology is the neurovascular unit (NVU) [5,6]. Understanding the functionality of the NVU is key to a more efficient approach to stroke. The NVU encompasses the endothelium, pericytes, astrocytes, microglia and neurons. These cellular elements of different embryological origin are anatomically and functionally connected to ensure the correct pairing of metabolic supply/catabolic discharge and CNS activity [5]. Oligodendrocytes are not classically involved in the NVU, although recently they have been shown to actively respond to ischemic insult [7]. The CNS vasculature consists of a specialized form of endothelium with tight junctions (TJs) that regulate osmotic exchanges between the blood and parenchyma [8], the so-called blood–brain barrier (BBB). CNS tissue structure is completed by non-cellular elements such as the basal membrane and other specialized forms such as the neuronal extracellular matrix (NECM), playing a central role in physiological conditions and pathological modifications [9,10].

Injury of the NVU induces the immediate disruption of the BBB, not limited to the ischemic area, with edema of the nearby tissue. Hemostatic and inflammatory responses are helpful in self-limiting acute damage, however, it may lead to an amplification of cellular loss and consequent long-term functional damage, particularly considering the massive necrosis induced by inflammatory responses. In the white matter, it causes further damage to myelin, thus propagating further CNS tissue damage via a vicious cycle mechanism [6,9]. Acute damage is caused by the loss of oxygen and nutrient supply, driven by vessel occlusion (e.g., atheroma), rupture or thromboses (due to endothelial dysfunction or embolization). Neuronal loss within the ictal core is the direct consequence of the acute energy failure. However, the permanent injury of the neuronal networks is mainly caused by the maladaptive reparative processes, the end product of a predominantly metabolic correlative damage of the NVU and NECM with the involvement of glial cells and the immune system [6,11]. Indeed, the inflammatory reaction following the injury within the CNS could be highly disruptive and sustained, with the infiltration of macrophages to the site of injury surrounded by astroglial reactive cells [12]

In this review, we propose evidence for cell matrix and NVU interactions in the pathogenesis of stroke and its sequelae. The experimental data call for the creation of protein–protein interaction (PPI) maps which define the importance of both structural (Figure 1) and sequential coupling (Figure 2 and Figure 3) of cellular and extracellular matrix pathways. The maps are designed using the string database platform [13] and are based on 38 selected proteins obtained through critical analysis of the literature. The papers used to create the structural PPI map are presented considering the logical structure of this review and the main experimental studies are presented in Table 1. The sequential coupling is obtained considering stroke pathophysiological data and the observed co-expression of proteins presented in Figure 2. This systems biology approach is essential to pave the way for an improvement in acute therapy and in the rehabilitation process.
ijms-21-07554-t001_Table 1Table 1Synoptic table of the main representative experimental studies analyzed to select 38 proteins used as input for the string database (Figure 1 and Figure 2). NIH: neuro-immune hemostasis, MMPs: matrix metalloproteinases, ADAMTS: a disintegrin and metalloproteinase with thrombospondin motifs, TIMP: tissue inhibitor of metalloproteinases, CSPGs: chondroitin sulfate proteoglycan, HSPGs: heparan sulfate proteoglycan, TnC: tenascin-C, TnR: tenascin-R, tPA: tissue plasminogen activator, GFAP: glial fibrillary acidic protein, ITGAM: integrin subunit alpha M, HMGB1: high-mobility group box 1 protein, TLR: Toll-like receptor, NFκB: nuclear factor-κB.Research PaperAnalyzed PathwaysSelected ProteinsModelQiu J et al., 2010 [14]Alarmins—MMPs—NIHHMGB1, TLR4, MMP9Mouse, in vivo and in vitroGu BJ and Wiley JS, 2006 [15]Alarmins—MMPs—NIHMMP9, P2X7R, TIMP-1Human, in vitroGao, H et al., 2011 [16]Alarmins—NIHHMGB1, ITGAM, NFkBMouse, in vitroChoi MS et al., 2010 [17]Alarmins—MMPs—NIHMMP9, P2YRRat, in vitroManaenko A et al., 2010 [18]Alarmins—NIHHSP70, IL1, TNFα, collagenMouse, in vivoMalik R et al., 2018 [19]NIH—MMPsNOS, COL4AHuman, clinicalClausen BH et al., 2008 [20]NIH—Glial activationIL1, TNFα, ITGAMMouse, in vivoBotchkina GI et al., 1997 [21]NIH—NeurotrophinsTNFα, p75NTRRat, in vivoAtangana E et al., 2017 [22]NIH—Glial activationP-selectin, Iba1, ITGAMMice, in vivoWeisenburger-Lile D et al., 2019 [23]NIH—Glial activationNeutrophil elastase, ITGAMHuman, clinicalStubbe T et al., 2013 [24]NIH—Glial activationITGAM, Iba1Mouse, in vivoChoucry AM et al., 2019 [25]NIH—Neurotrophinsp75NTR, TrkARat, in vivoCandelario-jalil E et al., 2011 [26]NIH—MMPsMMP2, MMP9Human, clinicalCheng T et al., 2006 [27]NIH—MMPsMMP2, MMP9, PAR1, Thrombin, tPA, NFkBMouse in vivo, Human in vitrodel Zoppo GJ et al., 2012 [28]NIH—MMPs—Glial activationMMP2, MMP9, GFAP, ITGAM, COL4A, HSPGsNonHuman Primate in vivo, mouse in vitroMishiro K et al., 2012 [29]NIH—MMPsMMP9, tPAMouse, in vivoChen W et al., 2009 [30]MMPs—Glial activationMMP2, MMP9, TIMP1, TIMP2, GFAPRat, in vivoYe H et al., 2015 [31]NIH—Glial activationS100BHuman, meta-analysisMaysami S et al., 2015 [32]NIH—Glial activationCXCL1, CCL3, Iba1Mice, in vivoQuattromani MJ et al., 2018a,b [33,34]NIH—MMPsMMP2, MMP9, ADAMTS4, TIMP1, tPA, CSPGs, TnC, TnRRat and Mouse, in vivoMatsumoto H et al., 2008 [35]NIH—Glial activationIba1, NG2, ITGAMRat, in vivoCarmichael ST et al., 2005 [36]MMPsCSPGsRat, in vivo
Figure 1Extracellular/intracellular network analysis within the damaged tissue (string-db.org platform) for protein–protein interactions (PPIs). The proteins related to stroke pathophysiology are clustered and PPIs are defined. This map is difficult to use for the design of further experiments since the interference with any node will apply to every edge at the same time, an atypical phenomenon in biological systems. The map is based on a critical analysis of recent literature (see Table 1 for selected experimental studies). The database query with 38 proteins showed 203 edges with an average node degree of 10.7 (expected number of edges 35, average local clustering coefficient 0.575; PPI enrichment *p*-value < 10^−16^). For further details, the map is accessible at this link: [37].
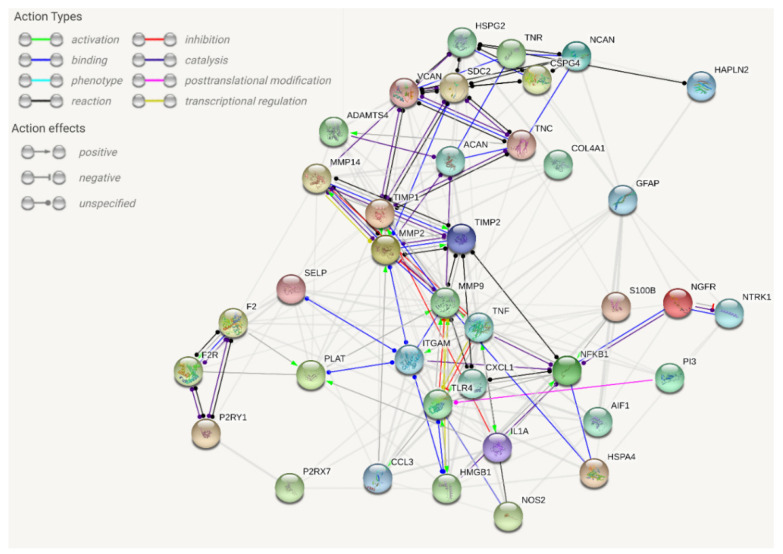

Figure 2Bidimensional matrix of RNA expression patterns and protein co-regulation. The 38 proteins proposed in the network map of Figure 1 are listed in a bidimensional matrix to show the co-expression scores (from 0 to 1 on the color visual scale) based on RNA expression patterns and protein co-regulation (string-db.org platform). The regulatory functions are subjected to a precise time-coupled expression. For further details, the matrix is accessible at this link: [38].
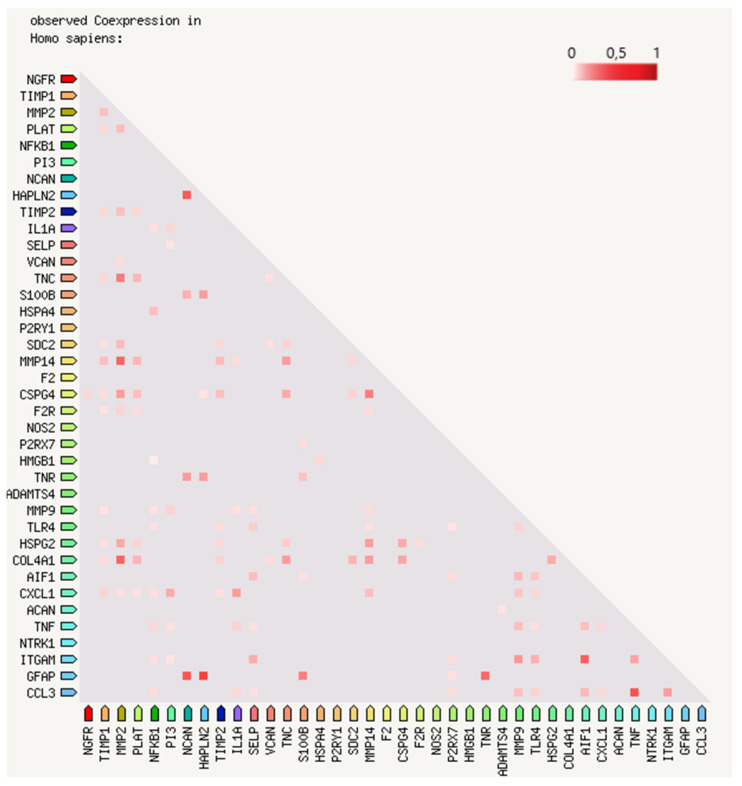

Figure 3Protein–protein interaction (PPI) maps generated considering the timescale of stroke. Clustering is dependent on time-specific activation, avoiding a direct interaction analysis (presented in Figure 1) and considering the collected literature data and co-expression matrix presented in Figure 2. (**A**) Alarmins are released and constitutive matrix metalloproteinase 2 (MMP2) and purinergic receptors are promptly activated following stroke. (**B**) The inflammasome activates the neuro-immune pathway of cytokines, adhesion molecules, protease receptors and inducible MMPs. (**C**) Leukocytes arrive hours after stroke, act with their enzymes and remain for several days and, while glial activation proceeds, a novel extracellular matrix (NECM) is secreted and reactive gliosis regulates the neurotrophin concentration and receptors.
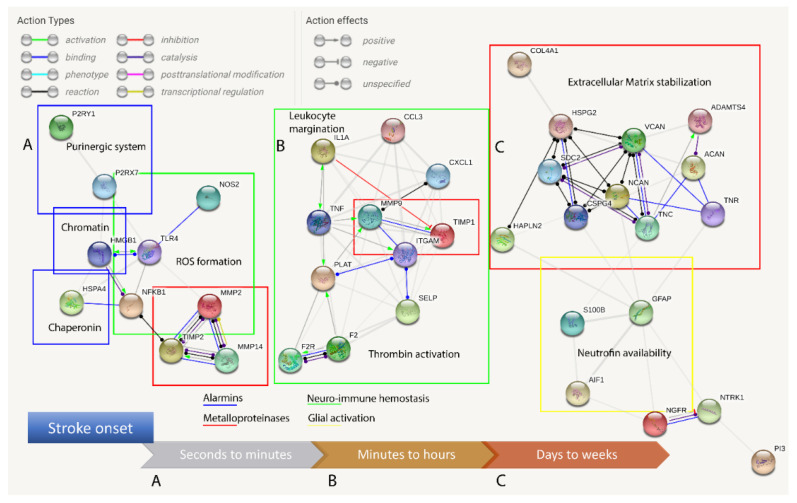


## 2. The Brain Energy Failure in Stroke

The interruption of regular perfusion causes an irreversible metabolic imbalance in the territories of the corresponding vasculature, which ultimately causes cell death. The extension and the nature of the vascular insult (e.g., small/large vessel disease, ischemic or hemorrhagic strokes) lead to a variety of parenchymal changes in the CNS, such as damage to cellular and extracellular elements in the ischemic core (i.e., cavitation), as well as a progressive reduction of the CNS tissue with spots of scar tissue and brain atrophy [39]. In the early phases of post-ischemic damage, typically within the first two weeks (before three months, i.e., standard chronic onset timepoint), the majority of patients with clinically diagnosed stroke will have well-defined cavitation at the ischemic core [40]. The cavity is surrounded and limited by glial cells that partially limit the area of damaged tissue (leukocyte infiltration, cellular debris, NECM and signaling molecules) [9,11] with long-lasting inflammation that could perdure more than three months according to injury models of myelinated tissues [41]. Myelin debris in particular should be removed to allow axonal regeneration and to reduce the inflammatory response, as it interferes with the crosstalk between the NVU (i.e., pericytes) and immune system (i.e., macrophages) during tissue repair [42,43]. The volume of the cavitation depends on the size of the afferent vessel and could range from 45 mL (medium sized cavity in a clinically determined stroke) to 150 mL in the case of a malignant middle cerebral artery occlusion [44]. If the damage is superficial, instead of the formation of a cavity, the area could be infiltrated by granulomatous tissue comprehending macrophages and blood vessels where glial cells are obliterated [41]. This macroscopic phenomenon is underlined by distinct molecular steps involving the NVU and NECM elements which interact with CNS resident and blood-derived immune cells [6].

## 3. Acute Ischemic Damage

The failure to meet the energy demand and counterbalance the osmotic gradient disrupts the organized membrane structures and cell death (necroptosis) [45], a mechanism different from apoptosis (caspase-mediated programmed cell death) or autophagy. The starving cells, lacking a trophic supply and oxygen, release signaling molecules during the death process, such as chaperonins, chromatin proteins and purinergic mediators [46,47]. The early modifications of the NECM generate degraded structural proteins. These molecules are the first to be released as damage-associated molecular patterns (DAMPs). These DAMPs are called alarmins and activate the innate immune response following an acute disruption of the BBB and before the margination of leukocytes [48]. Among the abovementioned molecules, the main factors seem to be high-mobility group box 1 protein (HMGB1), adenosine triphosphate (ATP) and heat-shock protein 70 (HSP70) [14,15,46,49]. Hyaluronic acid (HA), fibronectin and heparan sulfate (HSPGs) or chondroitin sulfate proteoglycans (CSPGs) are the main factors of the NECM [50]. These are endogenous molecules that can activate the immune response within minutes (Figure 3).

### 3.1. Necroptosis and Chromatin Exposure

HMGB1 is a nuclear protein with transcriptomics functions associated with chromatin structural proteins [51]. The release of the acetylated form of HMGB1 in the extracellular space, following ischemic injury, can activate the inflammatory response, binding to the Toll-like family receptors, particularly types 2 and 4 (TLR2/4) and the receptor for advanced glycation end-products (RAGE), expressed on the resident cells (neurons, astrocytes, microglia) with the secretion of matrix metalloproteinase 9 (MMP9) [14,47]. Indeed, the reshaping of the NECM is an early step, essential in the initial phases of neuroinflammation. Microglia, a scavenging component, express complement receptor 3 (CR3), an integrin α_M_β_2_, also known as macrophage antigen complex 1 (MAC1) which can bind HMGB1 and connect the immune response to cell adhesion and phagocytosis [6,16].

### 3.2. Purinergic System Activation

ATP has been extensively investigated in various damage models as an endogenous factor in maladaptive plasticity both in the acute and chronic rearrangement of the NECM and cellular elements (reactive gliosis), [15,17,52,53]. The activity of extracellular ATP in acute stroke could mediate the migration of microglia, assisted by the rapid secretion of MMP9 [15,17], reinforcing the HMGB1 cascade. The endothelium, neurons, glial cells and leukocytes express purinergic receptors (ionotropic P2Xs and metabotropic P2Ys) which account for the multimodal and rapid signaling function. Microglia–astrocyte signaling with ATP as a neurotransmitter is involved in the inflammatory CNS response with a prominent role of P2X4/7 and P2Y1/6/12 subtypes. ATP can tune the dead or alive response of oligodendrocytes, interacting with P2X7 or P2Y1 receptors, respectively [53]. The cation-selective opening of P2X receptors has an essential role in a faster response to the ligand and in a slower neuromodulatory response that needs to be further clarified [49]. The microglia–astrocyte crosstalk with ATP release and its metabolite adenosine diphosphate (ADP) has been reported in both glutamatergic excitotoxicity and astrocytic feed-forward potentiation in the early phases of reactive gliosis [11,54,55].

### 3.3. Chaperonine and Stress Response

HSP70 is a chaperonin expressed intracellularly and involved in protein folding or supporting the repair of misfolding processes. It has a role in the proteasome–ubiquitin catabolism of damaged protein [46]. HSP70 is part of a system identified to be stress inducible, with historical reference to heat shock, but currently known to be part of multiple stressor responses (e.g., ischemia, oxidative stress, misfolded protein accumulation) [56]. HSP70 is expressed either as a constitutive or inducible form. Recent studies have demonstrated the role of HSP70 in neuroprotection, considering ischemic and hemorrhagic stroke [46]. HSP70 seems to play a role in both intrinsic and extrinsic apoptotic pathways at multiple levels [46]. It has been shown to reduce vulnerability to specific ischemic insults in astrocytes, preserving higher intracellular ATP and glutathione levels with reduced reactive oxygen species (ROS) accumulation [57]. HSP70 exerts a bivalent role: (i) in anti-inflammatory intracellular pathways and (ii) by immunostimulating extracellular activity [46]. The overexpression of HSP70 reduced the production of tumor necrosis factor-α (TNFα) and IL1 and has been shown to attenuate BBB disruption in a model of hemorrhagic stroke [18,58]. The main HSP70-modulated intracellular systems are inducible nitric oxide synthase (iNOS) expression and nuclear factor-κB (NF-κB) activation [59]. An exonic polymorphism in NOS3 has been recently considered as a locus associated with stroke [19].

NF-κB deregulation probably caused the decreased expression of MMP9 in cultured astrocytes with HSP70 overexpression in an ischemic-like in vitro model [60]. In the same study, an MMP2 reduction was demonstrated, even if its regulation did not depend on NF-κB [60].

On the extracellular side, HSP70 (and other HSPs) can interact with TLR2/4, activate microglia and macrophages and promote lymphocyte differentiation [61,62,63].

These pathways are deeply connected with (i) neuro-immune hemostasis, (ii) MMP modulation and (iii) reactive gliosis.

## 4. Neuro-Immune Hemostasis

The essential molecules for transcript formation (HMGB1), cellular energetic metabolism (ATP) and protein-folding assistance (HSP70) constitute the endogenous pathology alarm molecules and damage control systems.

Establishing a network of such proteins in a systems biology model is essential to identify perturbations in metabolic supply, block the energy-consuming activities, save ATP, avoid ROS production and ultimately prompt a fast response to damage in the extracellular medium (Figure 1 and Figure 2). Whether the rescue system is effective or not, it depends on the well-timed resolution of the vascular insult. Nowadays, there are systemic and local treatments to guarantee good tissue reperfusion in ischemic stroke [64], however, the modulation of the intracellular–extracellular alarmins system could increase the time window for efficient therapy and improve the clinical outcome.

The activation of the coagulation cascade and the network involving clot formation, thrombin signaling, immune response and synaptic plasticity has been proposed in several neurological diseases [6]. During stroke, clotting dysfunction is part of the disease itself, with a prominent role in cell survival and the loss of the vital elements in the ischemic core.

### 4.1. Blood–Brain Barrier Failure

The damage to the BBB structure is the main harm to the NVU during acute stroke, it allows the permeation of blood elements, normally confined to the systemic circulation. These are constituted basically of plasma proteins and immune cells [65]. The DAMP response, on the other hand, produces in a few hours the second phase of the inflammatory process: the production of cytokines such as TNFα and IL1 [20,66]. The astrocytes of the NVU retract their podocytes, the basement membrane is modified by the release of MMPs and the production of cytokines promotes the margination of the first infiltrating leukocytes (neutrophils and monocytes) that act together with the microglia as host defenders.

The activated endothelial cells, in the area of vasogenic edema, express on their membrane cytokine receptors [21,67] and cell adhesion molecules (CAMs) of the immunoglobulin family, such as vascular (V)CAM-1 and platelet (P) and endothelial (E) selectins and integrins (e.g., MAC1), facilitating the rolling, margination and migration of leukocytes in the CNS [22,68].

Indeed, the expression of CAMs has been largely studied as a cell-based phenomenon, but more recent data have shown a direct involvement of NECM molecules, particularly laminin-8/10, fundamentally expressed in the basal lamina of the NVU. It has been shown that in mice lacking laminin-10, a facilitated infiltration of leukocytes with the loss of cadherin junctional support following TNFα administration occurred, while knocking out laminin-8 significantly reduced immune cell margination [69,70].

### 4.2. Perivascular Space Signaling

Perivascular spaces with low-density collagen type IV, a fundamental constituent of the basement membrane, and laminin are characterized by a prevalent neutrophil infiltration [71]. The mutation of the gene encoding for the α1 chain of collagen type IV, COL4A1, is responsible for a monogenic stroke syndrome (hemorrhagic and/or ischemic small vessel disease) [72], whereas an intron variant of the same gene has been associated with increased risk of multifactorial stroke [19].

The immune cell infiltration (hours to days) of the CNS utilizes both elastases and MMPs to further open the pathway to the second wave of leukocytes: the lymphocytes [23,73]. This process is self-limiting, with a modulatory function principally attributed to T cells, in the tardive phases (Figure 3) of tissue remodeling (weeks to a month timescale) [24,73].

The cytokine shift mediated by the regulatory T cells is typical of this phase, with the production of IL6/10, transforming growth factor β (TGFβ) and possibly neurotrophins such as brain-derived neurotrophic factor (BDNF) and nerve growth factor (NGF) [25]. The stabilization of the lesion is fundamental for neurological signs of partial recovery [74,75].

The schematic description of early to late phases of the immune response needs to be considered as overlapping phenomena playing a pivotal role in NECM remodeling and reactive gliosis (Figure 3) [6,11,76].

## 5. Matrix Metalloproteinase Modulation

The activation of MMPs is absolutely essential, as it begins during the early phases of stroke pathophysiology. The two main factors are MMP2 and MMP9. MMP2 is constitutively expressed and secreted as a zymogen and can be activated by the formation of a complex involving MMP14 (a membrane-type MMP), tissue inhibitor of metalloproteinase 2 (TIMP 2) and pro-MMP2 itself. MMP14 contains a hydrophobic C-terminal domain and its activity is restricted to the cell membrane [77]. This feature accounts for the spatial definition and limited activation of MMP2 [78]. MMP9 production, on the other hand, is induced during inflammation and its zymogen cleavage is mainly granted by MMP3, activated in neuroinflammatory responses [79]. TIMP1 is the natural inhibitor of MMP9 [80]. TIMP2 has a role as a sensor of damage and confinement of the injury, being able (if not complexed with MMP14) to inhibit MMP2 (Figure 2 and Figure 3) [79,81]. Similarly, the protease thrombin can activate the coagulatory cascade and, once complexed with thrombomodulin and endothelial protein C receptor (EPCR), can mediate the downregulation of coagulatory factors with activated protein C (aPC) [10]. Both MMP2 and MMP9 systems are associated with NVU dysfunction, mainly mediated by the breakdown of claudin-5, occludin and zonula occludens 1 (ZO-1) [26].

These functions are essential in the development of the CNS and in adult brain vascular remodeling following pressor insults, non-occlusive vascular lesions or microvascular recanalization [82,83,84]. Major ischemic injury causes massive damage to BBB permeability with tissue edema and reperfusion damage; these factors are independent of the extension of the lesion area beyond the volume of the ischemic core [85,86].

The thrombolytic treatment could enhance the activation of already produced MMPs [87,88]. Thrombin or recombinant tissue plasminogen activator (rt-PA)-mediated PAR1 activation contribute to NFκB-induced MMP9 expression [27]. The PAR1-mediated increment of MMP9, particularly through pericytes, could worsen the intracranial hemorrhage clinical outcome [89].

The efficacy of rt-PA has changed the perspective for ischemic stroke patients, however, understanding of the time schedule of NECM modifications could further improve the clinical outcome.

### Metalloproteinase Differential Activation

MMPs’ timely activation is fundamental in the pathophysiology of stroke. An improved neurological outcome has been shown in an MMP2-deficient mouse model, which presented a smaller hemorrhagic volume and transformation rate upon occlusion [90]. The MMP2 levels were not significantly increased in hemorrhagic transformation, and the infarct area was not affected by MMP2 knockout, which although apparently in contrast with the previous data, further demonstrates its role in the early phases of BBB failure, with a constitutive, scarcely inducible expression and is later replaced by other MMPs (i.e., inducible MMP9) [91].

MMP9 is mainly expressed by resident cells, such as neurons, astrocytes, microglia and endothelial cells [92]. Indeed, MMP9 seems to be involved in the late phases of parenchymal modifications. MMP9 is also expressed by infiltrating leukocytes, but these cells are not considered as the main source [28]. Within the first 24 h of stroke onset, MMP3 activation (via proteolytic cleavage) appears to affect MMP9 levels [79]. In vivo studies with a pan-inhibitor of MMPs (GM6001) showed interesting results, preserving the TJ proteins [29,30]. MMPs are essential in adaptive plasticity phenomena (neuroblast formation, revascularization, synaptic plasticity) [93,94]. MMPs’ selective and limited inhibition/activation seems to be the real translational target. Indeed, a potential detrimental effect of pan-MMP inhibition 7 days after cerebral ischemia has already been shown, with the blockage of physiologic reparative phenomena [93].

NVU components (mainly pericytes and endothelial cells) have been shown to be protected by another MMP inhibitor in a focal embolic ischemia model that avoids pericyte contraction and laminin degradation [95].

MMPs and a disintegrin and metalloproteinase with thrombospondin motifs (ADAMTS) are the major NECM modifiers (Figure 2) implicated in the regulatory functions both in health and diseases [9]. Their function is not restricted to the processing of the NECM, they modulate synaptic plasticity. It has been reported that the local inhibition of MMP9 through endogenous inhibitors, hypothermia and synthetic compounds could help in reducing the inflammatory process and improve the outcome in the acute phase [96]. In the subacute phase, MMP9 blockade could be harmful as it has been reported to play a role in the migration of neural progenitors of the subventricular zone (SVZ), suggesting a stimulating role for adult neurogenesis [94]. Moreover, ADAMTS4 has been considered useful in CSPG degradation and axonal regeneration [97].

## 6. Reactive Gliosis and NECM Deposition

The proliferative response of glial cells to an ischemic event is stimulated by cytokines [11]. An ongoing registry is evaluating cytokines in stroke, based on the previous finding of the S100B (an astrocyte marker) association with neurological outcome and infarct size [31,98]. The upregulation of glial fibrillary acidic protein (GFAP) expression has been generally used as a reactive astrocyte biomarker throughout various models of maladaptive CNS plasticity in CNS diseases [11,52,99,100]. Astroglial response functionally confines the damage with the recruitment of scavenger cells as peripheral macrophages to remove the debris of the acute damage, through the secretion of chemokines, classified based on the first two amino-terminal cysteine (C) sequences as adjacent (CC motif) or separated by another amino acid (CXC). CC ligand 3 (CCL3) and CXC ligand 1/2 (CXCL1/2) have been identified as the main chemokines involved (Figure 1) [32,101,102]. The damaged tissue is replaced by a newly formed NECM and by the proliferation of non-neuronal cells, particularly immune cells and glia (astrocytes and microglia) with the production of tenascins (Tn), proteoglycans and collagen (i.e., fibrous tissue) that rapidly aid tissue repair but definitively inhibit neural plasticity (potentially avoiding maladaptive rewiring) through the scar [103].

### 6.1. Astroglial Adaptive Plasticity

The subsequent propagation of the necrotic inflammatory response is limited by the surrounding protection of the glial cells. Modeling compromised scar formation showed increased neuronal dysfunction with combinatory death and demyelination in both ischemic and traumatic injury [104,105].

The glial signaling regulates the immune response, organizes the structural reshape of the NECM and avoids maladaptive neuronal plasticity. The leukocyte invasion is indeed regulated by CSPG and tenascin-C (TnC) expression [106,107]. Both these constituents and astrocytic-produced keratan sulfate proteoglycans (KSPGs) are regulatory molecules for neural plasticity and axonal outgrowth [108]. The second Tn expressed in the CNS (TnR) is instead expressed in the perineural nets (PNNs), representing another specialized form of NECM [92]. At this time, rigorous debate concerns if and how it could be beneficial for neural plasticity and limiting scar formation, and if it could play a role in rescuing post-lesional axonal sprouting. Even though targeting astrocytic proliferation and NECM production or NECM degradation has been proposed in various models of injury, [80,82,95,100,109] affirm that it is not always true that astrogliosis “per se” inhibits axonal sprouting and, moreover, it has been reported that astrogliosis has anti-inflammatory and anti-edema activity. The release of CSPGs and TnC may not be detrimental and may help axon regeneration and guidance, while the blockage of MMPs could be useful to increase neurotrophin levels, as demonstrated in injury models [11,100,110].

A recent model has been proposed focusing on the PNNs, a quadruple knockout mouse for brevican, neurocan and TnC/R [33]. It has shed light on NECM modifications in damaged tissue and in the functionally associated parenchyma. The results were controversial, with higher intrahemispheric connectivity and worse behavioral outcomes in the early phases of stroke, further contributing to the idea that a timely modulation is required [33].

### 6.2. CSPG/HSPG Expression

The NECM has a structural storage function for signaling molecules and growth factors [9,92]. During brain development, HSPGs are necessary for lineage differentiation, cell positioning and neural wiring [111,112]. An important role in the differential expression of CSPGs and HSPGs may provide dynamic path-finding for axonal wiring, with HSPGs enhancing and CSPGs restricting the outgrowth [113].

The CSPGs overexpressed in the late phases of stroke are neurocan, brevican, phosphacan and NG2 [35,114]. The latter is the marker of oligodendrocyte precursor cells (OPCs), which are a putative functional reservoir of multipotent cells, involved in the remyelination of damaged circuitry and supporting axonal outgrowth as the only non-neuronal cell that can form synapses with neurons [115]. OPCs contribute to NECM remodeling by upregulating versican in the damaged area [35]. CSPGs in the PNNs far from the lesion are reduced as a result of the plastic remodeling of the entire system following the injury [36].

A CSPG/HSPG ratio, similar to the developmental processes, has been observed in ischemic models and could be potentially targeted in order to modify neuronal plasticity [113]. The physiological role of the scar could be to inhibit the neural plasticity from the perilesional area into the ischemic core with a certain grade of synaptic plasticity in the ischemic penumbra and neighboring intact tissue [116]. The reactive microglia, astrocytes and leukocytes persist in the ischemic core and are responsible for the structure of the stroke-damaged tissue, including the NECM remodeling. However, the data available were totally produced, except for a few, in rodent animal models typically used in stroke research. We must consider that these animals show very little white matter in their hemispheres, therefore, a vascular injury such as obstruction of the middle cerebral artery leads to gray matter injury, with very little myelin involvement and self-limiting inflammation. To date, despite the growing amount of data, the correct understanding of the role of each element in the clinical outcome following vascular damage is very poor. The systems biology approach could provide designs for conceptual maps with the simultaneous analysis of multiple factors (Figure 1). The benefit of a new methodological approach aimed at assessing the overall molecular networks involved in a specific disease is needed to overcome the old attitude limited to a single cellular or structural element as the key to understanding the pathophysiology of a disease [117]. The therapeutic approach to these models could allow us to overcome the inhibition/enhancement dualism, focused on a single path (valid for monogenic diseases), as this strategy is very poor if applied to the multifactorial pathophysiology of sporadic stroke (Figure 2).

## 7. Designing the Network: The Time Variable

The modulation of the NECM should be investigated considering its cellular components and how these molecules act as intracellular signal transducers. To the same extent, it is impossible to investigate diseases clearly showing NVU impairment without analyzing its components and blood-derived infiltrating cells. All these factors have their prominent roles on the stage and are mutually influenced by a specific proximity and temporal activation/inhibition (Figure 3). The structural network (Figure 1) needs to be improved, adding the time variable to the same hub connections (Figure 3). In these networks, no targets are defined to block maladaptive phenomena and/or improve adaptive plasticity. The validation of the available data requires adequate experiments based on well-designed networks. These must consider the progression of the injury site and the timing of the reparative processes.

The scaffold of the NECM is mainly composed of HA, proteoglycans, collagen and Tns and, although many of these elements act as signaling molecules in precise phases (development, injury), they are expressed as constitutive elements in the normal adult brain, with physiological turnover but consistent stability [92]. Signaling factors with gradient expression or short-range efficacy are embedded among these functional nets. Indeed, many of these signaling molecules are strictly connected to the abovementioned structural elements and their regulators are found in the same environment [92].

## 8. Future Perspectives and Therapies

Novel experiments proposed on the basis of the designed rational networks could be helpful to elucidate the role of the NECM in the pathophysiology of stroke. Another limitation to overcome is the current knowledge of both ischemic and hemorrhagic stroke founded on various models (atherogenic, embolic, hypertensive) without discrimination of the involved pathways. BBB failure, on the other hand, could be used to redirect pharmacological treatment to selectively deliver anti-inflammatory drugs to the damaged and edematous tissue.

### 8.1. Metalloproteinases and Proteoglycans

There is one ongoing double-blind, placebo-controlled clinical trial evaluating minocycline (as an MMP9 inhibitor) and molecular hydrogen (an antioxidant) in stroke [118]. Their protective role could prevent ischemia/reperfusion injury. Moreover, minocycline has been showed to be effective in inhibiting poly(ADP-ribose) polymerase. This trial is a proof of concept for the results of a study that showed a possible role of minocycline in lowering MMP9 plasma levels following rt-PA treatment of stroke [119].

Another clinical trial [120] will also consider MMP9 levels as a possible predictive factor for hemorrhagic transformation of ischemic stroke treated with rt-PA.

Efforts to translate the data on NECM remodeling in stroke recovery is represented by the use of HSPG mimetics that, administered intravenously in an acute ischemic stroke model, demonstrated a good spatial localization and within 6 h of the event, reduced functional deficits and increased plasticity phenomena (neo-angiogenesis, neurogenesis). An open study, the first in humans, to assess the clinical safety of an HSPG mimetic (OTR4132) in anterior circulation acute ischemic stroke of patients re-vascularized with both rt-PA or endovascular thrombectomy, will be conducted [121].

### 8.2. Engineering Bioscaffolds

The inflammation post trauma or post stroke needs to be the first contained/eliminated issue before an implant is considered. However, the chance to replace the lost NECM using engineered biomaterial has been developed with a top-down approach of modified natural bioscaffolds to achieve a promising bottom-up hydrogel synthesis [122]. There are various technical issues to overcome to realize a translational success in this field, although there are promising results in animal models [122]. In particular, experimental data are based on rodent models of stroke and are limited to brain tissue (with higher gray/white matter ratio than humans) and could be reconsidered to favor nervous tissue richer in white matter, although this has other limitations [123,124].

One of the main biological challenges is represented by the intrinsic connectivity of the CNS. The replacement of novel neurons, infiltrating the bioscaffold, needs to be coupled with a robust newly formed and functional network. The restoration of complex cognitive functions can be achieved with the correct coupling of axonal growth and synapse formation both in local circuitry and with long-distance connections. Hydrogels with HA and vascular endothelial growth factor (VEGF) seem to be able to replicate angiogenesis and axonal networks in the infarct area, similar to the contralateral hemisphere [122]. Studies on axonogenesis, substance deliveries and cell influence on the newly formed tissue have been considered [125,126,127], but a lack of knowledge on the intricate network of biological, behavioral and experience-dependent processes halts the organization into a functional brain tissue. The major risk is the mere substitution of scar tissue with an organized but useless, or even detrimental, cell/matrix mass. Finally, the role of the environment is crucial for the development of a functional network, as already demonstrated by the impact of early physiotherapy in post-stroke recovery that could partly act on the remodeling of the NECM. It has been shown that an enriched environment could decrease PNNs around neurons and modulate the transcription levels of NECM-modifying proteins in the somatosensory cortex [34].

## 9. Conclusions

The treatment of stroke has so far been focused on limiting acute damage at the onset and securing the hemodynamic stabilization of the patient to reduce the loss of CNS tissue.

This approach, with rt-PA and mechanical thrombectomy approved for acute ischemic stroke, represented a major contribution and, together with early rehabilitation, significantly improved clinical outcomes. However, this success is limited to selected patients with rescuable brain tissue (within a time window or with significant ischemic core/penumbra mismatch). Hemorrhagic stroke without coagulatory diseases or aneurysmatic ruptures can be clinically modified solely with the strict control of blood pressure values. The actual paradigm is indeed to avoid further damage. We are unaware of how to affect disease progression depending on these variables in the post-acute phase. The focus of further research should be on the cellular capability to reduce the neuroinflammatory phenomena and actively reshape the matrix in early reperfusion and beyond. For instance, a patient treated with rt-PA could benefit from the reduction of constitutive MMP (e.g., MMP2) activation or the blockage of the thrombin-mediated PAR1 pathway. Mechanical thrombectomy, on the other hand, could assist in reducing the prominent BBB failure secondary to large vessel diseases, focusing on endothelial activation and leukocyte margination. These pathways, together with induced MMP activity (e.g., MMP9), could be partially shared with hemorrhagic stroke. These speculations may not actually be proven and may need a systems biology approach to better understand the adaptive reparative mechanisms and try to improve the sequalae with an active approach. MMP and cytokine interference, biomaterial scaffolds, growth signals and an enriched environment are slowly being translated into clinical trials.

The therapeutic benefits, however, may be flawed by the incorrect modulation of proteins without the understanding of the complex structural and temporal phases of their adaptive functioning. This could be the reason for the knockout or pan-inhibitory models that have so far provided contradictory results. All the elements analyzed, cellular and molecular, are affected by a single constant problem, the damage to the NVU. This results in an energy deficit that alters the behavior of all components of the CNS, as reported for neurodegenerative diseases [122]. Defining a model that allows an energy rescue of the NVU and the NECM is the real therapeutic goal.

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
