# Peer review of "Roadmap for Stroke: Challenging the Role of the Neuronal Extracellular Matrix"

_ijms, 2020, doi:10.3390/ijms21207554_

Round 1

Reviewer 1 Report

The manuscript is well-written and the results document important therapeutic activity of a number of proteins.

Minor revisions are suggested below.

  1. Figures 1 and 3 is not good for publication. Legends «Action Types» and «Action effects» are hard to see.
  2. Would like to see a more detailed description of the networking methodology in Figures 1 and 3

Author Response

Reviewer 1

Comments and Suggestions for Authors

The manuscript is well-written and the results document important therapeutic activity of a number of proteins.

Minor revisions are suggested below.

  1. Figures 1 and 3 is not good for publication. Legends «Action Types» and «Action effects» are hard to see. We improved the Figures based on your appreciated comment to make the text more readable.

  1. Would like to see a more detailed description of the networking methodology in Figures 1 and 3

We clarified the network, citing the database used to create the PPI map (string-db.org). Furthermore, we added the hyperlink where the map and the graph could be analyzed in detail for Fig. 1 and Fig. 2. We specified how Fig. 3 was obtained based on Fig.1 and Fig.2 and finally designed Table1 to explain the experimental studies that allowed us to choose these proteins to create the network.

Reviewer 2 Report

Imjs-920265

REVISIONS.

ABSTRACT.

Abstract vaguely indicates that stroke is a vascular event leading to a chronic disease involving metalloproteinases and reparative processes.  The usefulness of this abstract is low to anybody looking for a good review paper on pathogenesis of the disease called stroke.

The main lacking point, induction of destructive inflammation by the vascular event in stroke is not even mentioned.

INTRODUCTION.

The authors introduce important mechanisms of stroke but of peripheral significance to the disease.  In a confusing fashion. 

The authors avoided discussing the effect of ischemia-induced locally massive necrosis in a white matter of the brain that leads to the most severe and prolonged inflammation following a vascular incident (infarct) anywhere else in the body.  Therefore, the authors attempt at review of stroke as a disease is obsolete and does not address its pathogenesis.

Given the above, complex molecular relationships presented by the authors are difficult to interpret and probably meaningless until stroke-induced severe inflammation is considered.

SELECTED COMMENTS.

  • BBB damage is not localized to the brain tissue surrounding the area of necrosis/inflammation in this paper. Since it appears that vasogenic edema persist as long (beyond 16 weeks post-injury) as the inflammatory cells in the area of inflammation, this is an important mechanism in the pathogenesis of stroke for at least 2 main reasons: (1) Persistence of vasogenic edema due to leaky BBB, (2) damaged BBB opens the route for tissue delivery of anti-inflammatory drugs that can cross it (some will cross it some will not).
  • There is no such thing as “glial scar” after spinal cord injury (SCI), traumatic brain injury (TBI) or stroke. Of two types of inflammation induced by trauma or stroke; (a) deep lesions are converted into a cavity of injury (COI) where a severe macrophage-rich inflammatory infiltration is sequestered by astrogliosis forming a wall at the brain-COI interface and (b) superficial or connected to wide superficial injury lesions infiltrated by severe granulomatous infiltration involving macrophages, fibroblasts and blood vessels called arachnoiditis (in SCI but appropriate in TBI and stroke) which progresses into a scar devoid of neural cells including glial cells.  This scar becomes walled off from the surrounding CNS by a thick wall of astrogliosis.  The notion of “glial scar” as a part of CNS is therefore inappropriate.
  • Since arachnoiditis and resulting scar are not part of the CNS, it is a hostile, impossible barrier for CNS axonal regeneration. The same applies to the COI which fills with water by astrocytic syncitial systems from the areas with persistent vasogenic edema.  Axons do not swim.
  • Given severity and destructive power of macrophage-rich inflammation induced by the white matter injury in SCI, TBI and stroke any cellular and/or synthetic implants to serve as a bridge for axonal regeneration across the injury site require inhibition and elimination of inflammation. Then there is the inhibitory effect of myelin sheaths past the area of injury.  The sheaths need to be removed in a safe fashion (without inducing severe inflammation) in areas of CNS targeted for axonal re-growth.

A number of references indicate rodent models of stroke.  Since the rodent (mouse, rat) brain hemispheres are particularly poor in the white matter the injury is the gray matter injury that initiates limited inflammatory response that is self-limiting.  Contrary to that injury initiated by stroke in the human brain involves very often massive damage to the white matter.  Therefore, the rodent brain is an inappropriate model of stroke.  An animal model with a much larger, complex brain and rich in the subcortical white matter is needed, a pig brain injury may serve this purpose.  Alternatively, injury to a myelin-rich rodent spinal cord should be considered to study pathogenesis and involved molecular mechanisms of stroke.

Author Response

Reviewer 2

REVISIONS.

ABSTRACT.

Abstract vaguely indicates that stroke is a vascular event leading to a chronic disease involving metalloproteinases and reparative processes.  The usefulness of this abstract is low to anybody looking for a good review paper on pathogenesis of the disease called stroke.

The main lacking point, induction of destructive inflammation by the vascular event in stroke is not even mentioned.

We thank you the reviewer and following his critics we improved the section, particularly stressing, as suggested, the inflammatory response.

INTRODUCTION.

The authors introduce important mechanisms of stroke but of peripheral significance to the disease.  In a confusing fashion. 

The authors avoided discussing the effect of ischemia-induced locally massive necrosis in a white matter of the brain that leads to the most severe and prolonged inflammation following a vascular incident (infarct) anywhere else in the body.  Therefore, the authors attempt at review of stroke as a disease is obsolete and does not address its pathogenesis.

Given the above, complex molecular relationships presented by the authors are difficult to interpret and probably meaningless until stroke-induced severe inflammation is considered.

As carefully suggested, the inflammatory response has been properly highlighted in this section and endothelial activation, BBB failure, leukocyte margination following the ischemic injury are part of the complex molecular map presented. The attempt is to elucidate the molecular pathways using a sequential approach, we tried to clarify our purpose. Thank you for the valuable comment.

SELECTED COMMENTS.

  • BBB damage is not localized to the brain tissue surrounding the area of necrosis/inflammation in this paper. Since it appears that vasogenic edema persist as long (beyond 16 weeks post-injury) as the inflammatory cells in the area of inflammation, this is an important mechanism in the pathogenesis of stroke for at least 2 main reasons: (1) Persistence of vasogenic edema due to leaky BBB, (2) damaged BBB opens the route for tissue delivery of anti-inflammatory drugs that can cross it (some will cross it some will not).
  • We absolutely agree with the reviewer that BBB damage and failure is not limited to the ischemic area, we made it clearer where could be misunderstood. In particular, we stated: “Major ischemic injury cause massive damage of BBB permeability with tissue edema and reperfusion damage; these factors are independent of the extension of the lesion area beyond the volume of the ischemic core [68,69]” Further, we added a sentence to underlie the noteworthy influence of BBB opening as drug delivery route.
  • There is no such thing as “glial scar” after spinal cord injury (SCI), traumatic brain injury (TBI) or stroke. Of two types of inflammation induced by trauma or stroke; (a) deep lesions are converted into a cavity of injury (COI) where a severe macrophage-rich inflammatory infiltration is sequestered by astrogliosis forming a wall at the brain-COI interface and (b) superficial or connected to wide superficial injury lesions infiltrated by severe granulomatous infiltration involving macrophages, fibroblasts and blood vessels called arachnoiditis (in SCI but appropriate in TBI and stroke) which progresses into a scar devoid of neural cells including glial cells.  This scar becomes walled off from the surrounding CNS by a thick wall of astrogliosis.  The notion of “glial scar” as a part of CNS is therefore inappropriate.
  • We absolutely agree with the reviewer and eluded the term “glial scar” although it is widely used in literature to describe the astrogliosis surrounding the area of the damage. SCI and TBI are not part of this review, however, as properly suggested, we mentioned the inflammatory response related to myelinated tissue damage as a particularly devastating process and stressed the difference between deep and superficial lesion of the CNS, as carefully recommended.
  • Since arachnoiditis and resulting scar are not part of the CNS, it is a hostile, impossible barrier for CNS axonal regeneration. The same applies to the COI which fills with water by astrocytic syncytial systems from the areas with persistent vasogenic edema.  Axons do not swim.
  • Once the injury site has formed an organized structure with active inflammation and is, as mentioned, hostile to axonal regeneration there is no need to try to enhance adaptive phenomena. We made changes as suggested to better clarify this point to avoid misleading sentences.
  • Given severity and destructive power of macrophage-rich inflammation induced by the white matter injury in SCI, TBI and stroke any cellular and/or synthetic implants to serve as a bridge for axonal regeneration across the injury site require inhibition and elimination of inflammation. Then there is the inhibitory effect of myelin sheaths past the area of injury.  The sheaths need to be removed in a safe fashion (without inducing severe inflammation) in areas of CNS targeted for axonal re-growth.
  • We thank the reviewer for this suitable consideration, therefore we discussed the potential role of myelin debris in the inhibition of axonal regeneration and rewiring.

A number of references indicate rodent models of stroke.  Since the rodent (mouse, rat) brain hemispheres are particularly poor in the white matter the injury is the gray matter injury that initiates limited inflammatory response that is self-limiting.  Contrary to that injury initiated by stroke in the human brain involves very often massive damage to the white matter.  Therefore, the rodent brain is an inappropriate model of stroke.  An animal model with a much larger, complex brain and rich in the subcortical white matter is needed, a pig brain injury may serve this purpose.  Alternatively, injury to a myelin-rich rodent spinal cord should be considered to study pathogenesis and involved molecular mechanisms of stroke.

We absolutely agree with the reviewer, the rodent model is generally insufficient to replicate human diseases affecting brain tissue. The same is true for degenerative diseases. However using pig or primate model or rodent spinal cord has several limitations in terms of the technical procedures for many laboratories (especially the pig model) and for differences in cellular morphology, cytoarchitecture, and structural organization for the sc. Moreover, we have to consider how is difficult to test a  more complex behavior. Therefore, we appreciate and mentioned your suggestion, this would increase and promote further studies.

Reviewer 3 Report

The review paper provides an emphasis on the importance of a system biology approach in understanding the role of neuronal extracellular matrix and the treatment of acute stroke. My comments are the following.

  1. The resolution of the figures (especially, Figure 1 and Figure 3) should be improved. Please explain in detail how the graph was obtained. Perhaps, citation is necessary for the figures. 
  2. I think it will be helpful if the authors provide a Table that highlights a few of important research articles, which address the benefit of the system biology approaches in acute stroke therapy.   
  3. There are several treatment options (e.g., IV thombolysis, endovascular thrombectomy) and independent variables (e.g., symptom onset to reperfusion time, infarct location) that affect functional outcome in acute ischemic stroke.  It may be useful if the authors provide the disease progressions in cellular levels after a given treatment scenario and how they differ depending on the treatment options and conditions. 

Author Response

Reviewer 3

Comments and Suggestions for Authors

The review paper provides an emphasis on the importance of a system biology approach in understanding the role of neuronal extracellular matrix and the treatment of acute stroke. My comments are the following.

  1. The resolution of the figures (especially, Figure 1 and Figure 3) should be improved. Please explain in detail how the graph was obtained. Perhaps, citation is necessary for the figures. 

The image has been re-submitted in high resolution, to improve the images, based on your appreciated comment we rewritten the legend, magnified the string-database output and rearranged the text. We cited the database used to create the PPI map (string-db.org), to elucidate the data we added the hyperlink where the map and the graph could be analyzed in detail for Fig. 1 and Fig. 2. We also express in Table1 some of the experimental studies that allowed us to choose these proteins for the network.

  1. I think it will be helpful if the authors provide a Table that highlights a few of important research articles, which address the benefit of the system biology approaches in acute stroke therapy.   

Systems biology approach to stroke is quite an explorative field, we highlighted in a table the reports that allowed us to create the PPI map and were designed to address the pathophysiology of stroke accounting multiple pathways.

  1. There are several treatment options (e.g., IV thrombolysis, endovascular thrombectomy) and independent variables (e.g., symptom onset to reperfusion time, infarct location) that affect functional outcome in acute ischemic stroke.  It may be useful if the authors provide the disease progressions in cellular levels after a given treatment scenario and how they differ depending on the treatment options and conditions. 

We mentioned the differences between the approved treatments for stroke and patient characteristics. To address your appreciated comment we pointed out in the Conclusion some of the pathways that could be differently modulated based on actual treatment options and clinical scenarios

Round 2

Reviewer 2 Report

I am encouraged by the authors' willingness to address the fundamental issues that have dramatically changed our understanding of the disease initiated by stroke.  The paper will benefit.

i think that the following comments may be helpful to clarify some of these issues in this review paper.

Lines 68-69:     “…particularly considering the massive necrosis induced by inflammatory responses in the white matter.”

Reviewer:        Actually, a vascular event resulting in ischemia results in massive necrosis.  Inflammatory infiltration by leukocytes is in response to necrosis and then if it is in the white mater, it causes further damage to myelin thus propagating further CNS tissue damage via a mechanism of vicious cycle.

Lines 359-362: Definitively, it could be beneficial for neural plasticity to limit scar formation, rescuing post-lesional axonal sprouting. Targeting astrocytic proliferation and NECM production or NECM degradation has been proposed in various models of injury.

Reviewer:        The above statement is obsolete.  By scar formation authors likely mean “glial scar” which should be glial hypertrophy + hyperplasia = astrogliosis.  Astrogliosis does not inhibit axonal plasticity = sprouting or axonal regeneration = regrowth.  Also, astrogliosis, a hallmark of CNS tissue reaction to injury and inflammation appears to be anti-inflammatory and also helps removing excess water from areas with vasogenic edema [refs. 12, 15].  Therefore, astrogliosis is of beneficial effect in stroke and in restoration of homeostasis and inhibition of astrogliosis appears to be counterproductive.  Promotion of astrogliosis with its anti-inflammatory and anti-edema activity seems like a way to go when designing therapeutic strategies in management of stroke.  The authors may not want to discuss this issue in this paper because it is novel and based on systematic histologic analysis only. But astrogliosis inhibiting axonal sprouting or regeneration is simply not true and evidence for astrogliosis having anti-inflammatory (water soluble) and anti-edema activity is rather compelling. These are fundamental issues of CNS tissue reaction to stroke and neurotrauma and they need to be addressed for the sake of effective neuroprotective and anti-edema therapies.

Lines 393-394: The reactive microglia, astrocytes, and leukocytes persist in the 393 ischemic core and are responsible for the structure of the stroke-damaged tissue, including the NECM 394 remodeling.

Reviewer:        Perhaps the persistence of live neural cells may be possible in gray matter stroke.  However, we are more concerned with ischemic lesion initiated by a vascular accident in the white matter, very abundant in the human brain.  The white matter stroke or injury resulting in local necrosis does not contain neural cells but myelin-rich necrotic debris, red blood cells, few degenerating blood vessels, few scattered neutrophils.  It then becomes infiltrated by macrophages by day 3.  Large amounts of damaged myelin act as a very powerful immunogen and initiated a very severe inflammation (the most severe in the body after trauma).  This may not happen in the gray matter injury.  The real issue here is that rodent animal models typically used in stroke research have very little white matter in their hemispheres, therefore, a vascular injury such as obstruction of the middle cerebral artery leads to gray matter injury, with very little myelin involvement and self-limiting inflammation. 

In their reply, authors suggest that an animal model such as a pig or monkey is problematic.  I agree; a simian model of stroke is expensive and ethically difficult.  Pig model of stroke (I don’t think it has been developed) may be less expensive and less ethically challenging but difficult nevertheless.  Ironically, the spinal cord injury model in a rodent (rat) may be of value here (model of stroke) because it is rich in myelin and it is small therefore easy to manipulate pharmacologically.  I am not suggesting this debate should be included in the present paper, just clarifying an important issue for the benefit of the authors and the readers.

8.2. Engineering bioscaffolds

Reviewer:        This chapter may confusing.  The authors are right to indicate that rodent brain has little white matter as opposed to the real (human) need.  Again, avoiding the issue of inflammation initiated by the injury prior to the implantation of a hydrogel is a fundamental flaw.  A white matter injury-induced inflammation will simply destroy the implanted material and reject it by forming fluid filled cavities surrounding it.  Therefore, post-trauma or post-stroke inflammation needs to be first contained/eliminated before the implant is considered.  Also, I am not sure about the value of the implant to reconstitute a gray matter injury = reconstitute neurons and attending glia.  My impression is that a theoretical purpose of an implanted scaffold/hydrogel is to bridge a lesion that severed axons and lead to axonal regeneration across such a bridge.  There are several issues fundamental to the axonal regeneration and the white matter regeneration in such an approach.  None of them addressed yet, therefore hard to review.

Author Response

Comments and Suggestions for Authors

I am encouraged by the authors' willingness to address the fundamental issues that have dramatically changed our understanding of the disease initiated by stroke.  The paper will benefit.

I think that the following comments may be helpful to clarify some of these issues in this review paper.

Lines 68-69:     “…particularly considering the massive necrosis induced by inflammatory responses in the white matter.”

Reviewer:        Actually, a vascular event resulting in ischemia results in massive necrosis.  Inflammatory infiltration by leukocytes is in response to necrosis and then if it is in the white mater, it causes further damage to myelin thus propagating further CNS tissue damage via a mechanism of a vicious cycle.

We thank you the reviewer for the suggestion to better elucidate the pathogenetic mechanism and added in the text. However, necrotic cell death induced by the ischemic insult is mainly due to the fatal failure of oxygen and glucose supply. Cell debris released with necrosis into the tissue trigger leukocytes infiltration, which starts clearing the debris and amplifies the inflammation sustained by the early activation of microglia and astrocytes in the ischemic penumbra (Wang et. al, 2016, DOI:10.1016/j.pneurobio.2016.04.005; Puig B et al., 2018 10.3390/ijms19092834. ). 

Lines 359-362: Definitively, it could be beneficial for neural plasticity to limit scar formation, rescuing post-lesional axonal sprouting. Targeting astrocytic proliferation and NECM production or NECM degradation has been proposed in various models of injury.

Reviewer:        The above statement is obsolete.  By scar formation authors likely mean “glial scar” which should be glial hypertrophy + hyperplasia = astrogliosis.  Astrogliosis does not inhibit axonal plasticity = sprouting or axonal regeneration = regrowth.  Also, astrogliosis, a hallmark of CNS tissue reaction to injury and inflammation appears to be anti-inflammatory and also helps to remove excess water from areas with vasogenic edema [refs. 12, 15].  Therefore, astrogliosis is of beneficial effect in stroke and in the restoration of homeostasis, and inhibition of astrogliosis appears to be counterproductive.  The promotion of astrogliosis with its anti-inflammatory and anti-edema activity seems like a way to go when designing therapeutic strategies in the management of stroke.  The authors may not want to discuss this issue in this paper because it is novel and based on systematic histologic analysis only. But astrogliosis inhibiting axonal sprouting or regeneration is simply not true and evidence for astrogliosis having anti-inflammatory (water-soluble) and anti-edema activity is rather compelling. These are fundamental issues of CNS tissue reaction to stroke and neurotrauma and they need to be addressed for the sake of effective neuroprotective and anti-edema therapies.

Also in this case we thank you the reviewer for theories and studies proposed on the authentic and current debate concerning astrogliosis. We analyzed these complex phenomena in very different models in the past 20 years, in vitro as in vivo, “sadly” in rodents! We agree with the term obsolete because it is absolutely obsolete to consider astrocyte as a unique type of cell. We agree with the reviewer that we have to move on and define carefully as suggested for decades by several seminal papers published by the beloved Ben Barres, the type of astrocyte we are studying, probably this is the matter. However, we changed the statement in the manuscript. Studies show that while astrocytes have long been viewed to play scaffolding and supportive roles for neurons, activated astrocytes may also be detrimental in the ischemic brain (Kim JY, 2016, /doi.org/10.5607/en.2016.25.5.241). Time matters. Astrocytes limit brain damage, reduce neuroinflammation, and are critical for BBB reconstruction and maintaining CNS homeostasis in the acute stages of ischemic stroke. In the chronic stages, these cells hinder functional recovery and axon regeneration, also suggesting that they participate in the progression of stroke (Xu S, 2020, doi: 10.3389/fimmu.2020.00294).

Lines 393-394: The reactive microglia, astrocytes, and leukocytes persist in the 393 ischemic core and are responsible for the structure of the stroke-damaged tissue, including the NECM 394 remodeling.

Reviewer:        Perhaps the persistence of live neural cells may be possible in gray matter stroke.  However, we are more concerned with ischemic lesions initiated by a vascular accident in the white matter, very abundant in the human brain.  The white matter stroke or injury resulting in local necrosis does not contain neural cells but myelin-rich necrotic debris, red blood cells, few degenerating blood vessels, few scattered neutrophils.  It then becomes infiltrated by macrophages by day 3.  Large amounts of damaged myelin act as a very powerful immunogen and initiated a very severe inflammation (the most severe in the body after trauma).  This may not happen in the gray matter injury.  The real issue here is that rodent animal models typically used in stroke research have very little white matter in their hemispheres, therefore, a vascular injury such as obstruction of the middle cerebral artery leads to gray matter injury, with very little myelin involvement and self-limiting inflammation. 

We discussed this issue in the previous report with the reviewer. Several reports imply different contributions of neutrophils and microglia/macrophages to white matter injury after ischemic stroke. The distinct localization of activated microglia/macrophages implies complex signals that regulate their migration toward and infiltration of damaged white matter in rodent models (Moxon-Emre I. et al,2010). Multiple models are necessary to replicate the complete spectrum of human white matter stroke lesions. However, distinct histological changes as focal edema, demyelination, axonal damage, loss of oligodendrocytes, and local activation of astrocytes and microglia have been found within ischemic white matter rarefaction (Malee S. F et al, 2006, https://doi.org/10.1161/01.STR.0000221308.94473.14; Sozmen EG et al, 2012, https://doi.org/10.1007/s13311-012-0106-0). HOWEVER, WE CONSIDER AND CHANGES THE TEXT.

In their reply, authors suggest that an animal model such as a pig or monkey is problematic.  I agree; a simian model of stroke is expensive and ethically difficult.  Pig model of stroke (I don’t think it has been developed) may be less expensive and less ethically challenging but difficult nevertheless.  Ironically, the spinal cord injury model in a rodent (rat) may be of value here (model of stroke) because it is rich in myelin and it is small therefore easy to manipulate pharmacologically.  I am not suggesting this debate should be included in the present paper, just clarifying an important issue for the benefit of the authors and the readers.

WE THANK YOU, CORDIALLY, THE REVIEWER AND WE INTRODUCED THIS CRITICAL CONCERN IN THE LAST VERSION: LINES 377-380

8.2. Engineering bioscaffolds

Reviewer:        This chapter may confusing.  The authors are right to indicate that rodent brain has little white matter as opposed to the real (human) need.  Again, avoiding the issue of inflammation initiated by the injury prior to the implantation of a hydrogel is a fundamental flaw.  A white matter injury-induced inflammation will simply destroy the implanted material and reject it by forming fluid filled cavities surrounding it.  Therefore, post-trauma or post-stroke inflammation needs to be first contained/eliminated before the implant is considered.  Also, I am not sure about the value of the implant to reconstitute a gray matter injury = reconstitute neurons and attending glia.  My impression is that a theoretical purpose of an implanted scaffold/hydrogel is to bridge a lesion that severed axons and lead to axonal regeneration across such a bridge.  There are several issues fundamental to the axonal regeneration and the white matter regeneration

We share with the several concerns presented by the reviewer on this matter, and we inserted as incipit of the subchapter, an alert statement. Absolutely, there are several issues fundamental to the axonal regeneration and network connectivity following injury. Moreover, recent advancements in the fabrication of artificial bio-interfaces have yielded an enhanced understanding of the phenomenahttps://doi.org/10.3390/cells9051074 ; https://doi.org/10.1088/1741-2552/abb9c0; https://doi.org/10.1038/s41563-020-0703-y), encouraging us to review the new perspectives in this field. Considering that the biomedical implant should be performed following a therapy aiming at containing/eliminating the inflammation, we propose ideas regarding a controlled composition of the extracellular matrix to guide the processes of functional recovery post-stroke.